# The Dynamic Impacts of Monetary Policy Uncertainty Shocks

**Ahmed Kamara** [1] and **Niraj P. Koirala** [2,*]

1   Department of Decision Sciences and Economics, Texas A&M University—Corpus Christi, 6300 Ocean Drive, Corpus Christi, TX 78412, USA

2   Department of Economics and Statistics, California State University Los Angeles, 5154 State University Dr, Los Angeles, CA 90032, USA

*   Correspondence: nkoiral2@calstatela.edu

**Abstract:** This paper assesses whether the impact of monetary policy uncertainty on the U.S. economy has changed over time. Estimating a Time-Varying Parameter Vector Autoregressive model on U.S. data from 1985Q1 to 2022Q3, we find that uncertainty shocks have larger negative effects on output during the COVID-19 recession than during other periods. However, financial market variables, such as stock prices and dividends, responded more significantly to uncertainty shocks during the Asian crisis of the late 1990s, the IT bubble of the 2000s, and the Great Recession. We then develop a Dynamic Stochastic General Equilibrium model with monetary policy uncertainty. Based on the calibrated model, we conduct several counterfactual exercises to demonstrate that the effects of uncertainty shocks depend on the state of the economy, which is consistent with the empirical evidence. These findings provide new insights into the time-varying nature of the impact of economic uncertainty.

**Keywords:** DSGE; TVP-VAR; uncertainty shocks; COVID-19

## 1. Introduction

The financial crisis of 2008/09 led to a renewed interest in the relationship between uncertainty and economic variables. This led to a significant amount of research in this area including Bloom (2009); Basu and Bundick (2017); Beetsma and Giuliodori (2012); Mumtaz and Theoridoris (2018). The consensus is that uncertainty shocks have negative impacts on the economy. While this is significant, the focus of these studies have primarily been on economic uncertainty in general. They are also mostly VAR-based analyses that focus on a single economic event, such as the Great Recession, except for Beetsma and Giuliodori (2012); Mumtaz and Theoridoris (2018) who attempted to study the dynamic effects of uncertainty across different events. Mumtaz and Theoridoris (2018), in particular, argue that the negative impact of uncertainty shocks on the economy are declining over time. However, the timing of their research precludes the recent crisis that was triggered by the COVID-19 pandemic.

In the wake of the pandemic, the Federal Reserve Bank (Fed) engaged in a number of policy actions to salvage the economy, the latest of which is the ongoing interest rate hikes to combat the historic rising prices which appear to be persistent. This has led to a lot of uncertainty in the monetary policy environment with regards to how far the Fed is willing to go and how long the interest rate hikes would last. In this paper, we use a Time-Varying Parameter Vector Autoregression (TVP-VAR) model to assess the economic impact of monetary policy uncertainty (MPU) across three major economic events including the IT bubble of the 2000s, the Great Recession, and the COVID-19 recession. In particular, we assess whether the impact of such monetary policy uncertainty shocks varies across these major economic events or not, using US data from 1985Q1 to 2022Q3. Our proxy for uncertainty is the news-based MPU index constructed by Husted et al. (2017). This

index captures the degree of uncertainty that the public perceives about Federal Reserve policy actions and their consequences. Moreover, our framework incorporates time-varying parameters that allow for estimating time-varying response of economic variables to such MPU shocks.

Our findings indicate that monetary policy uncertainty shocks have had a larger impact on output during the COVID-19 recession compared to the other major economic events. However, the impact of MPU shock on financial market variables, such as stock prices and dividends, were more significant during the Great Recession, the Asian crisis, and the IT bubble of the 2000s compared to that of the COVID-19 recession. These findings provide new insights into the belief that the impact of uncertainty shocks are declining over time (see, for instance, Mumtaz and Theoridoris (2018)).

In order to assess the role of different channels through which MPU shocks manifest in the economy, we simulate a Dynamic Stochastic General Equilibrium (DSGE) model with asset markets. In the DSGE model framework, we attempt to study uncertainty shocks by incorporating a new policy strategy published by FOMC in light of COVID-19. According to the strategy, monetary policy would respond to a moving average of deviations in inflation from the FOMC long-term goal of 2%. Similarly, in contrast to the standard Taylor rule, the new policy would target the moving average of output from its steady state. The DSGE simulation appears to capture some of the amplifying effects of uncertainty shocks on macroeconomic and financial variables. Furthermore, the simulation results suggest that a change in the Fed's stance on inflation and wage and price rigidities might affect the dynamism of uncertainty shocks over time.

The paper contributes to the literature in a number of ways. First, the use of the TVP-VAR model allows us to determine whether the effects of monetary policy uncertainty shocks on the US economy have increased or decreased over time. Second, the findings that stock prices and output have suffered their largest impacts at different times suggest that the impact of uncertainty shocks might be asymmetric across different parts of the economy. Third, by using a new monetary policy formulation in the DSGE that is based on the new FOMC strategy, this paper attempts to add new insights into the broader topic of state dependent uncertainty shocks.

The paper is organized as follows: Section 2 explains the empirical framework and its results. In Section 3, we present the dynamic impacts of uncertainty shocks using a Dynamic Stochastic General Equilibrium (DSGE) model and a Monte Carlo experiment based on the DSGE framework. In Section 4, we conclude the paper with a discussion.

*Uncertainty in the Literature*

Interest in the study of uncertainty has been evolving since the work of Bernanke (1983). After a revived interest in the study of the macroeconomic effects of uncertainty shocks by Bloom (2009), several other works have explored different mechanisms through which uncertainty shock propagate in the economy (see, for example, Born and Pfeifer 2014; Carriero et al. 2015; Leduc and Liu 2016; Mumtaz and Surico et al. 2018). Some of the mechanisms for the transmission of uncertainty shocks include the Oi–Hartman–Abel affects, real options effects, precautionary savings, and countercyclical markup channels.

The so-called Oi–Hartman–Abel effect (due to Oi 1961; Hartman 1972; Abel 1983) assumes that if profits are convex in demand or costs, then shocks to uncertainty about demand or cost increases expected benefits. Therefore, through the Oi–Hartman–Abel channel, uncertainty shocks are expansionary in nature. However, Born and Pfeifer (2014); Basu and Bundick (2017) find that the presence of sticky prices opens the possibility for 'inverse-Oi–Hartman–Abel-effect'. In the sticky-price model, firms choose higher markups following an increase in uncertainty, thereby decreasing output. To this end, uncertainty shocks are contractionary through the countercyclical markup channel even when profits are convex in demand or costs.

The real options channel of uncertainty can be originally attributed to Bernanke (1983). The idea behind this mechanism is that firms are likely to wait for some time in making

decisions about new investments and hiring until there is a resolution of uncertainty shocks. In the case of the precautionary savings channel, uncertainty shocks increase consumers' desire to save more, thereby decreasing consumers' total expenditure (see Bansal and Yaron 2004). While a rise in precautionary savings might increase output in the long-term , the short-term effects are contractionary. Moreover, in the case of a highly open small economy, some of the savings will flow to foreign economies, which dampens both domestic demand and future growth (Bernanke 1983).

Apart from the above-named channels, there have been recent attempts to explore the credit channel for the propagation of uncertainty shocks in the economy (see Aghion et al. 1999, Gilchrist et al. 2017; Bordo et al. 2016; Valencia 2017; Bianchi and Corugedo 2018; Choi et al. 2018; Brand et al. 2019). The goal is to see how the rise in uncertainty affects credit market conditions and how that transmits into the macroeconomy. The consensus is that uncertainty shocks lead to a decline in the supply of credit in the economy, thereby decreasing investment.

Although existing papers in the field explain the economic impacts of uncertainty shocks using different channels, they mostly use constant coefficient VAR models. These models succeed in estimating the stylized information about uncertainty impacts that the rise in future uncertainty brings adverse impacts in an economy through different channels. However, these papers do not explain the dynamic nature of uncertainty shocks in the US economy. This paper attempts to fill the gap in the literature by using a dynamic VAR model, which uses a Bayesian approach to estimate uncertainty shocks. The details of the model are explained in the upcoming section.

## 2. Empirical Framework

In building the empirical framework, we follow the approach developed by Chan et al. (2020). The empirical framework uses a Time-Varying Parameter Vector Autoregression (TVP-VAR) model. First, we present the model in the general form with p lags as follows:

$$y_t = a_t + A_{1t}y_{t-1} + \ldots\ldots\ldots + A_{pt}y_{t-p} + \epsilon_t, \tag{1}$$

where $\epsilon_t \sim N(0, \Sigma)$. Define $X_t \otimes [1, y'_{t-1}, \ldots., y'_{t-p}]$ and $\beta_t = vec([a_t, A_{1t}, \ldots\ldots, A_{pt}]')$ and rewrite the above system as

$$y_t = X_t\beta_t + \epsilon_t. \tag{2}$$

The time-varying parameters $\beta_t$ are assumed to evolve as a random walk,

$$\beta_t = \beta_{t-1} + u_t, \tag{3}$$

where $u_t \sim N(0, Q)$, and the initial conditions $\beta_0$ are treated as parameters. For simplicity, we assume that the covariance matrix $Q$ is diagonal, i.e., $Q = \text{diag}(q_1, \ldots, q_{kn})$, where k = np + 1 is the number of explanatory variables in each equation of the TVP-VAR. It is important to note that the TVP-VAR model is a state space model where Equation (1) works as a measurement equation, while Equation (3) is a state equation.

The following independent priors for $\Sigma$, $\beta_0$ and the diagonal elements of $Q$ are considered:

$$\Sigma \sim IW(v_0, S_0), \beta_0 \sim N(a_0, B_0), q_i \sim IG(v_{0,qi}, S_{0,qi}), \tag{4}$$

where *IW* means inverse Wishart, *N* refers to normal distribution, and *IG* stands for independent inverse gammas. Finally, a Bayesian approach is used for estimating the TVP-VAR system. Bayesian inference in this type of model requires a Gibbs sampler to estimate the system. The Gibbs sampler for the TVP-VAR model can be summarized as follows. First, some initial values of $\beta^{(0)}, \Sigma^{(0)}, Q^{(0)}$ and $\beta_0^{(0)}$ are chosen. Then, the following steps are repeated from z = 1 to Z:

- Draw $\beta^{(z)} \sim (\beta|y, \Sigma^{(z-1)}, Q^{(z-1)}, \beta_0^{(z-1)})$ (multivariate normal);

- Draw $\Sigma^{(z)} \sim (\Sigma|y, \beta^{(z)}, Q^{(z-1)}, \beta_0^{(z-1)})$ (inverse Wishart);
- Draw $Q^{(z)} \sim (Q|y, \beta^{(z)}, \Sigma^{(z)}, \beta_0^{(z-1)})$ (independent inverse gammas);
- Draw $\beta_0^{(z)} \sim (\beta_0|y, \beta^{(z)}, \Sigma^{(z)}, Q^{(z)})$ (multivariate normal).

### 2.1. Variables in the System and Identification Strategy

There are four variables in the TVP-VAR model. The data for the analysis spans from the first quarter of 1985 to the third quarter of 2022. The ordering of the variables in the TVP-VAR system is as follows:

$$(TVP - VAR) = \begin{bmatrix} \text{Proxy of Uncertainty} \\ \text{GDP} \\ \text{Stock Price} \\ \text{Dividend} \end{bmatrix}$$

As a measure of uncertainty, we use the Husted–Rogers–Sun Monetary Policy Uncertainty Index for the United States. The index was constructed by Husted et al. (2017) by searching for keywords related to monetary policy uncertainty in *The New York Times*, the *Washington Post* and the *Wall Street Journal*. Figure 1 shows the time-series of the uncertainty proxy. The cyclical property of the uncertainty proxy makes it a very good fit for measuring uncertainty in the economy. As shown in the figure, the uncertainty proxy is highly countercyclical. The data show that the proxy rises significantly during economic downturns represented by shaded regions, with noticeable spikes during the crises of the 1990s, 2008/09 and the most recent and ongoing global health crisis precipitated by COVID-19. Since the major aim of this paper is to study the dynamic effects of monetary policy on the US economy, we believe that the chosen proxy makes perfect sense for our cause. Besides the monetary policy uncertainty index, we collect data on GDP, stock prices, and dividends. We use the S&P Composite price for stock prices. The data on dividends are obtained from the website of Prof. Robert Shiller. Nominal variables are adjusted for inflation. Except for the proxy of uncertainty, all other variables are log differenced. In a robustness check, we also use monthly data. In addition, we also employ the Michigan Consumer Uncertainty as the proxy of uncertainty. The data are collected from 1985Q1 to 2022Q3 due to the lack of monetary policy uncertainty data before 1985Q1. Following the literature on uncertainty shocks, we use Cholesky decomposition to estimate time-dependent impulse response functions from MCMC draws. Cholesky decomposition assumes that an innovation in uncertainty has impacts on non-uncertainty variables. However, any rise in non-uncertainty variables will not have impacts on the uncertainty proxy. We use the lag length of two during the analysis.

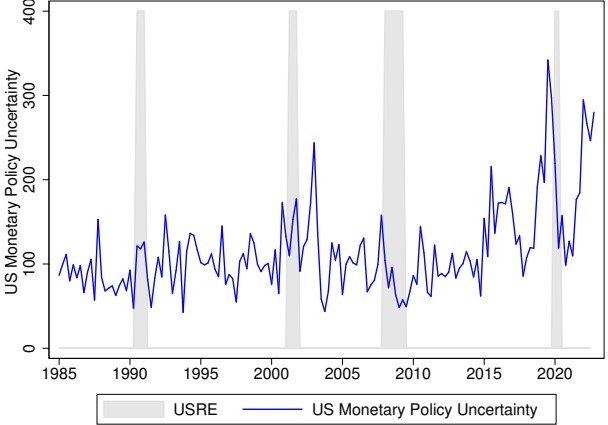

**Figure 1.** US monetary policy uncertainty. Notes: Figure 1 shows the time-series of US monetary policy uncertainty. In the figure, the shaded regions represent recessions in the US economy.

### 2.2. Impulse Response to Uncertainty Shocks

Figure 2 below shows 3D impulse response functions.

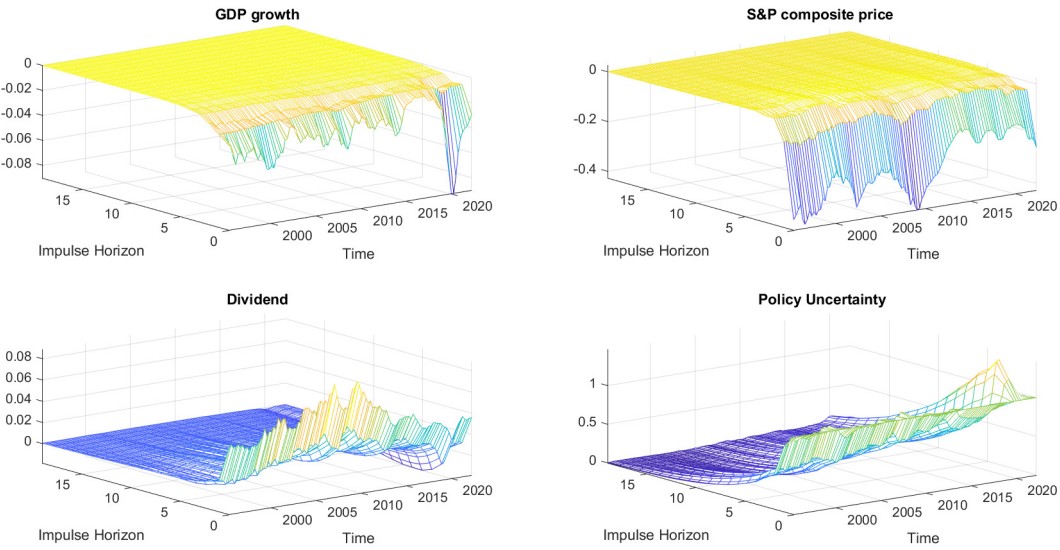

**Figure 2.** Impulse response functions to uncertainty shocks. Notes: Figure 2 shows one standard deviation to uncertainty.

Figure 2 shows the time-varying impulse response of variables to uncertainty shocks. The figure shows that uncertainty shocks have impacts of different magnitudes over the sample periods. Consider the first figure in the first row, which shows the response of the GDP growth rate to uncertainty shocks. The figure shows that uncertainty shock decreases output by 0.02% during the recession of 2001. However, output declines by about 0.03% as a result of uncertainty shocks during the Great Recession. Finally, the figure shows that the effects of uncertainty shocks on output growth were larger during the COVID-19 induced recession. Specifically, the contribution of policy uncertainty on output decline comes out to be around 0.08% decline in output growth in early 2020 and 2021. The second figure in the first row shows the response of stock prices to uncertainty innovations. The figure shows that the response of stock prices to uncertainty shocks is negative and the extent of the impact varies over time. For instance, uncertainty shocks reduced stock prices by 0.28% during the stock market downturn of 2002. Furthermore, the figure suggests that uncertainty shocks reduced stock prices by about 0.4% and 0.22% during the Great Recession and the COVID-19 period, respectively.

The response of dividends to uncertainty shock is also negative, and the impact varies over time. For example, the first figure in the second row suggests that policy uncertainty increased dividends by about 0.04% during the Great Recession of 2008/09 and in early 2020 when COVID-19 started ravaging the global economy. Lastly, uncertainty innovation increased policy uncertainty, while the increase in uncertainty increased by a larger percentage from 2020 to 2021, probably as a result of COVID-19.

To sum up, policy uncertainty has negative impacts on output, stock prices, and dividends in the US economy. Impacts of uncertainties on the GDP growth rate are in the range of −0.02 to −0.08%, while that on stock prices range from −0.2 to −0.4%. These findings suggest that policy uncertainty shocks might have negligible impacts on US business cycle fluctuation. These findings are in line with the literature (see, for example, Born and Pfeifer 2014).

For further understanding of dynamic impacts of uncertainty shocks, we calculated impulse response functions of the selected variables at three different periods: 1997Q4, 2008Q4 and 2015Q3. These three dates were chosen for different reasons. For instance, the response of variables to uncertainty shocks in 1997Q4 might help us understand the effects

of uncertainty during the Asian crisis, while 2008Q4 was chosen for studying uncertainty effects during the Great Recession of 2008/09. Lastly, 2015Q3 was chosen so that uncertainty shock contributions during the recovery phase can be studied. We could not estimate the dynamic responses for the pandemic years due to parameterization issues due to lack of sufficient data. Impulse responses in three different times are shown in Figure 3 below.

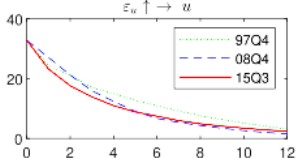 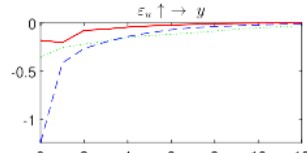 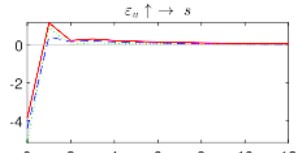 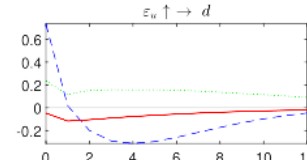

**Figure 3.** The Effects of Uncertainty Shock in 1997, 2008, and 2015. Notes: Figure 3 shows impulse response of variables to uncertainty shocks. - - indicates responses of variables to one standard deviation uncertainty shocks in 2015Q3. - - is response in 2008Q4 and the thin line is response in 2015Q3. In the figure, 'u' means uncertainty, 'y' is gdp growth, 's' is stock price return, and 'd' means dividend.

From Figure 3, we can see that uncertainty shocks have negative impacts on the GDP growth rate and stock prices and mixed impacts on dividends in all the three time periods under consideration. However, uncertainty impacts vary across time. For instance, GDP declined by larger percentage points during the Great Recession compared to the Asian crisis and the recovery phase of the Great Recession. The same holds true for stock price returns. Similarly, dividends suffered larger and persistent uncertainty impacts during the Great Recession. Additionally, the figure suggests that uncertainty impacts on dividends were less persistent during the stock market selloff of 2015–2016. Figure 3 once again suggests that uncertainty shocks might have different impacts on macroeconomic as well as financial variables.

*2.3. Robustness Checks*

In this subsection, we conduct a series of robustness checks. First, we replace the Monetary Policy Uncertainty Index (MPUI) with the Economic Policy Uncertainty Index (EPUI) from Baker et al. (2015) as an indicator of policy uncertainty. The new proxy is calculated from three types of underlying components: newspaper coverage of policy related to economic uncertainty, the number of federal tax code provisions set to expire, and the dispersion between between individual forecaster predictions about future values of the Consumer Price Index, federal expenditures, and local expenditures in constructing uncertainty about future values of economic variables. Uncertainty impacts using the new indicator are reported in Figure 4.

As in the baseline model, Figure 4 suggests that uncertainty impacts on output growth were largest during the pandemic times. While stock prices fell by the largest amount during the Great Recession of 2008/09 followed by uncertainty impacts during the dot com bust of the late1990s. As in the baseline model, dividends exhibited positive movement opposite to the response of stock returns.

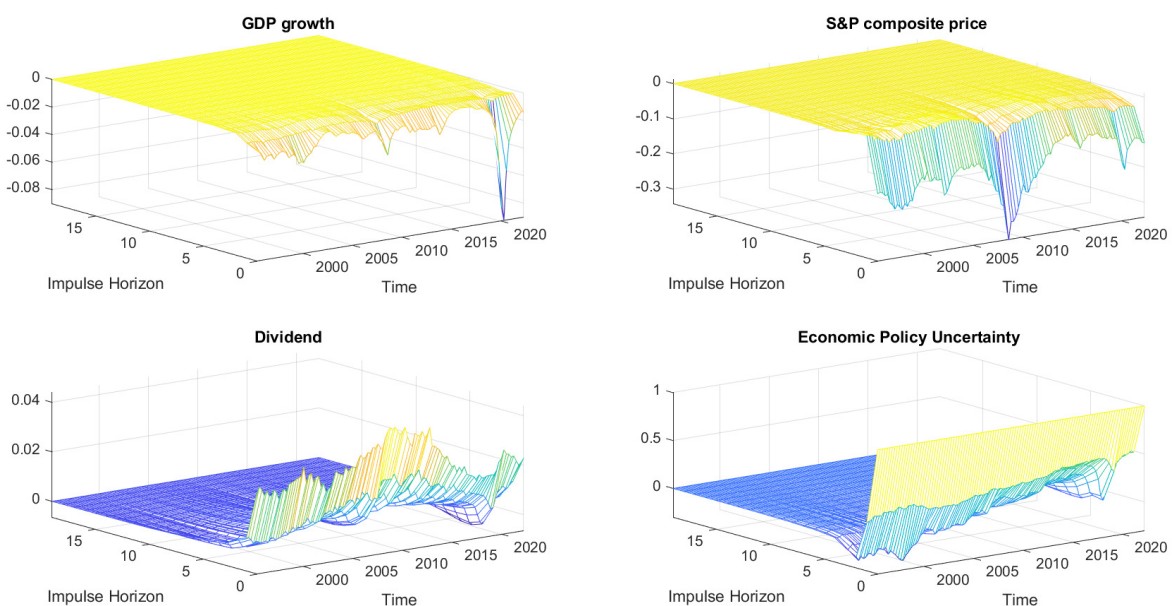

**Figure 4.** Impulse Response functions to Economic Policy Uncertainty Index. Notes: Figure 4 shows responses to one standard deviation in shocks to EPUI.

Dynamic Impacts of Uncertainty Shocks Using Monthly Data

In this subsection, we replicate the baseline analysis using monthly data. When using the monthly frequency of data, we use the Industrial Production Index as a proxy of economic activity. Results of estimation of the TVP-VAR model with the Industrial Production Index in it are illustrated in Figure 5.

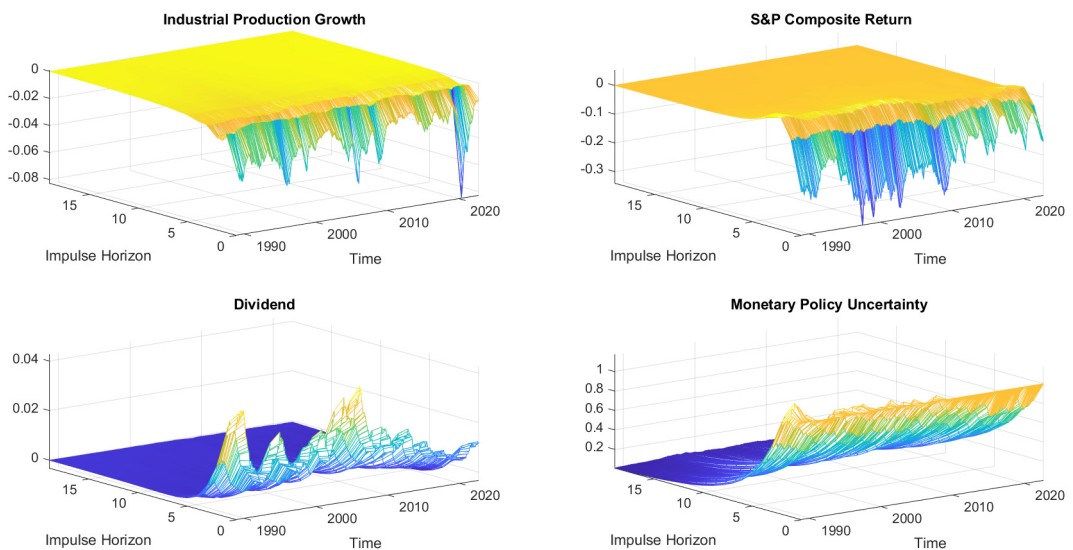

**Figure 5.** Impulse response functions to Economic Policy Uncertainty Index. Notes: Figure 5 shows dynamic responses to one standard deviation in shocks to MPUI.

Figure 5 shows the 3D impulse response functions of variables to increases in monetary policy uncertainty. As in the baseline model with quarterly data, the new model with monthly data suggests that policy shocks had the largest impacts during the pandemic years compared to any other previous times. Just like in the baseline model, the S&P

Composite Price return was affected the worst during the dot com bust of the late 1990s and the early 2000s, followed by that during the financial crisis.

In Appendix A, we show the TVP-VAR analysis results using another proxy of uncertainty, i.e., Michigan Consumer Uncertainty constructed by the University of Michigan based on a monthly nationwide survey of consumers across the country. Although the uncertainty variable mainly represents the uncertainty perception from the demand side, use of the variable might give us some idea about consumer perception about uncertainty due to policy issues among others.

## 3. A Monte Carlo Experiment: A DSGE Analysis

In this section, we study the role of different factors affecting the dynamic nature of uncertainty shocks using a Dynamic Stochastic General Equilibrium (DSGE) model. The model used in this paper is the one developed by Challe and Giannitsarou (2014). Following the uncertainty literature, we augmented this model with uncertainty in monetary policy. We also later amended the monetary policy rule to reflect the change in monetary policy stance of the Federal Reserve in the early days of COVID-19. Briefly, the model features the following sectors: (A) household sector, (B) intermediate goods market, (C) final goods market and (D) monetary policy. Each sector is explained in detail below:

### 3.1. Household Sector

There are $j$ number of households with $j \in [0, 1]$. In each period, a representative household supplies labor $N_{j,t}$ and derives utility from consumption $C_{j,t}$, where $C_t = \int_0^1 C_{j,t-1} dj$. The household, given the intertemporal discount factor, maximizes utility which is governed by the following function:

$$E_t \sum_{t=0}^{\infty} \beta^t \gamma_t \left\{ \frac{(C_{j,t} - bC_{j,t-1})^{1-\sigma}}{1-\sigma} - \chi \frac{N_{j,t}^{1+\nu}}{1+\nu} \right\} \tag{5}$$

where $\gamma_t$ is preference shifter, $\nu$ is the labor supply elasticity, and $\chi$ is the disutility of labor. Each household maximizes Equation (5) subject to the following budget constraint:

$$C_{j,t} + \frac{B_{j,t}}{P_t} + \int_0^1 V_{j,t}(h)Q_t(h)dh = \frac{W_{j,t}N_{j,t}}{P_t} + \frac{I_{t-1}B_{j,t-1}}{P_t} + \int_0^1 V_{j,t-1}(h)(Q_t(h) + D_t(h))dh, \tag{6}$$

where $P_t$ is the nominal price of goods, $W_{j,t}$ is the nominal wage of $j$ labor, $B_{j,t}$ and $V_{j,t}(h)$ denote the holdings of nominal bonds and shares of firms $h$ by household $j$ at the end of period $t$, respectively, $I_t$ is the gross interest on nominal bonds, and $Q_t(h)$ and $D_t(h)$ are the real price of a share of firm $h$ and dividend paid out by a firm $h$.

The Euler equations for bonds and shares $h \in [0, 1]$ are given by:

$$E_t \left[ \frac{M_{t,t+1} I_t}{\Pi_{t+1}} \right] = 1, \tag{7}$$

$$E_t [M_{t,t+1} R_{t+1}^e(h)] = 1, \tag{8}$$

where $\Pi_t = \frac{P_t}{P_{t-1}}$ is the gross inflation rate, and $M_{t,t+i} = \frac{\beta \lambda_{t+s} \gamma_{t+1}}{\gamma_t \lambda_t}$ is the stochastic discount factor for a payoff paid at date $t + i$, with $\lambda_t$ being marginal utility. $R_{t+1}^e$ is the ex post return on holding firm $h$'s shares from $t$ and $t + 1$. It is given by:

$$R_{t+1}^e(h) = \frac{(Q_{t+1}(h) + D_{t+1}(h))}{Q_t(h)}, \tag{9}$$

where $D_t(h)$ and $Q_t(h)$ are the stock dividend and trading price of firm $h$ at date $t$, respectively.

Wage Setting

Household *j* has monopolistic power over labor supply. More specifically, households' differentiated raw labor types are combined into homogeneous final labor by a competitive intermediary sector with production function:

$$N_t^d = \left( \int_0^1 N_t(j)^{\frac{\theta_w - 1}{\theta_w}} \right), \theta_w > 1, \tag{10}$$

where $N_t(j)$ is the supply of labor of j type, and $N_t^d$ is the economy wide demand for final labor by intermediate goods firms equal to $\int_0^1 N_t(h)dh$, where $N_t(h)$ is the labor demand by a firm (*h*). Cost minimization by labor intermediary and zero-profit condition:

$$\int_0^1 W_t(j)N_t(j)dj = W_t N_t^d \tag{11}$$

gives the labor demand for each labor type of *j* as follows:

$$N_t(j) = \left( \frac{W_t(j)}{W_t} \right)^{-\theta_w} N_t^d, \tag{12}$$

where

$$W_t = \left( \int_0^1 W_t(j)^{1-\theta_w} dj \right)^{\frac{1}{1-\theta_w}} \tag{13}$$

is the price of a final unit of labor. From Equation (12), labor supply can be written as:

$$N_t = \int_0^1 N_t(j)dj = \Delta_{w,t} N_t^d, \tag{14}$$

where

$$\Delta_{w,t} = \int_0^1 \left( \frac{W_t(j)}{W_t} \right)^{-\theta_w} dj \tag{15}$$

is the index of cross-household wage dispersion.

Household j sets the nominal wage so as to maximize utility given by Equation (5). There are nominal rigidities in the wage setting process. Every household resets its nominal wage optimally with probability $1 - \psi_w \in [0, 1]$ in every period and allows its previous wages $W_{t-1}(j)$ to grow at $\Pi_{w,t} = \frac{W_{t-1}}{W_{t-2}}$. The optimal wage for a household that can reset its wage is identical across households. Denoting optimal wage $W_t*$, it can be shown that the optimal wage satisfies the following situation:

$$E_t \sum_{t=0}^{\infty} \psi_w^i M_{t,t+i} \left[ \frac{W_t * J_{t,t+i}}{P_{t+i}} - \frac{\theta_w}{\theta_w - 1} \left( \frac{W_t * J_{t,t+i}}{W_{t+i}} \right)^{-\eta\theta_w} \frac{S_{t+i}}{\Delta_{w,t+i}^{\eta}} \right] \left( \frac{W_{t+i}}{W_t} \right)^{\theta_w} \frac{J_{t,t+i}^{-\theta_w} N_{t+i}}{\Delta_{w,t+i}} = 0, \tag{16}$$

In Equation (16), $J_{t+1}$ is the role of indexation for non-reset wages, and $S_t$ is the average marginal rate of substitution. From the constraint of the nominal wage adjustment in Equation (16), the wage rate for the final labor and wage indexation for raw labor can be written as:

$$W_t^{1-\theta_w} = \left[ (1 - \psi_w)(W_t*)^{1-\theta_w} + \psi_w (\Pi_{w,t-1} W_{t-1})^{1-\theta_w} \right], \tag{17}$$

$$\Delta_{w,t} = (1 - \psi_w) \left( \frac{W_t*}{W_t} \right)^{-\theta_w} + \psi_w \left( \frac{\Pi_{w,t-1}}{\Pi_{w,t}} \right)^{-\theta_w} \Delta_{w,t-1}. \tag{18}$$

### 3.2. Firms

#### 3.2.1. Final Good Producers

The demand schedule for final goods is given by the following equation:

$$Y_t(h) = \left(\frac{P_t(h)}{P_t}\right)^{-\theta_p} Y_t, \tag{19}$$

where $\theta_p$ is the price elasticity of demand, and $P_t(h)$ is the price of intermediate good $h$ and

$$P_t = \left(\int_0^1 P_t(h)^{1-\theta_p} dh\right)^{\frac{1}{1-\theta_p}} \tag{20}$$

is the nominal price of final goods. We also define the index for cross-firm nominal price dispersion as follows:

$$\Delta_{p,t} = \int_0^1 \left(\frac{P_t(h)}{P_t}\right)^{-\theta_p} dh. \tag{21}$$

#### 3.2.2. Intermediate Goods Producers

In our model, households hold firm shares and firms own capital stock. Therefore, firms decide the amount of capital to accumulate. Firms face Calvo price shocks; thus, firms having different histories of nominal prices will typically accumulate different levels of capital. The model assumes that firms accumulate capital not only from one period to the next but also trade a unit of capital within a period of time in a competitive market following the realization of Calvo price distribution. Specifically, any firm h saves a quantity of capital $K_t(h)$ at the end of time $t-1$ hoping that the price of capital stock in time $t$ will be $R_t^k$. After the idiosyncratic shocks are realized, a firm h may sell or buy additional capital at price, $R_t^k$, which results in the operational capital of $K_t\tilde{(}h)$. Furthermore, the model believes that the price of capital, $R_t^k$, adjusts every time so that a firm's total savings, $\int_0^1 K_t(h)dh = K_t$, equals total capital in use, $\int_0^1 K_t(\tilde{h})dh$.

The production function for firm h is given by:

$$Y_t(h) = Z_t \tilde{K}_t(h)^\alpha N_t(h)^{1-\alpha}, \tag{22}$$

where $Z_t$ is the total factor productivity,

The firms' budget constraint is written as:

$$D_t(h) + \Omega_t N_t(h) + R_t^k K_t\tilde{(}h) + X_t(h) = \frac{P_t(h)}{P_t} Y_t(h) + R_t^k K_t(h), \tag{23}$$

where $X_t(h)$ is investment and a capital reallocation of size $R_t^k(K_t\tilde{(}h) - K_t(h))$ takes place for firm $h$. Investment evolves according to the following equations:

$$K_{t+1}(h) = (1-\delta)K_t(h) + \left(1 - \tau\left(\frac{X_t(h)}{X(h)_{t-1}}\right)\right), \tag{24}$$

where $\delta \in (0,1)$ is the depreciation rate, and $\tau(.)$ is a capital adjustment cost, which has the following form:

$$\tau\left(\frac{X_t(h)}{X_{t-1}(h)}\right) = \frac{\varrho}{2}\left(\frac{X_t(h)}{X_{t-1}(h)} - 1\right)^2, \varrho > 0. \tag{25}$$

Firms maximize their values to the stakeholders, i.e., they choose $P_t(h)$, $X_t(h)$, $X_t\tilde{(}h)$, $K_t\tilde{(}h)$ and $N_t(h)$ to solve:

$$V(K_t(h), P_{t-1}(h), X_{t-1}(h), \mathfrak{C}_t(h), S_t) = max D_t(h) + E_t[M_{t,t+1}V(K_{t,t+1}(h), P_t(h), X_t(h), \mathfrak{C}_{t+1}(h), S_{t+1})], \tag{26}$$

where $\mathfrak{C}_t(h) = 1$ if the firm re-optimizes its selling prices in period $t$ and $\mathfrak{C}_t(h) = 0$ otherwise. Past investment $X_{t-1}(h)$ enters the current value function as a state variable due to its impact on the future adjustment costs.

Solving Equation (26) with respect to (22) and (25), the model has the following four equations:

$$\frac{K_t}{N_t} = \frac{\alpha}{1-\alpha} \frac{\frac{W_t}{P_t}}{\Delta_{w,t} R_t^k}, \tag{27}$$

$$K_{t+1} = (1-\delta)K_t + \left(1 - \tau\left(\frac{X_t}{X_{t-1}}\right)\right) \tag{28}$$

$$Q_t = E_t[M_{t+1}(R_{t+1}^k - (1-\delta)Q_{t+1}], \tag{29}$$

$$1 = Q_t\left[1 - \frac{\varrho}{2}\left(\frac{X_t}{X_{t-1}} - 1\right) - \varrho\left(\frac{X_t}{X_{t-1}} - 1\right)\frac{X_t}{X_{t-1}}\right] + E_t\left[M_{t+1}Q_{t+1}\varrho\left(\frac{X_{t+1}}{X_t} - 1\right)\right] \tag{30}$$

Equation (27) gives the optimal capital–labor ratio in terms of their relative prices. Equation (28) gives the evolution of capital in aggregate form. Equation (29) gives the current price of an additional unit of capital installed. Equation (30) is an expression for aggregate investment as a function of past investment and the marginal value of capital. Finally, the real marginal cost of the whole economy is calculated to be:

$$mc_t = \frac{1}{Z_t}\left(\frac{\frac{W_t}{P_t}}{1-\alpha}\right)^{1-\alpha}\left(\frac{R_t^k}{\alpha}\right)^{\alpha}. \tag{31}$$

Finally, the aggregate demand looks like:

$$D_t = Y_t - \frac{W_t}{P_t}\frac{N_t}{\Delta_{w,t}} - X_t \tag{32}$$

The price adjustment mechanism is assumed to be similar to that in Christiano et al. (2014). In every period, a firm is allowed to adjust its price with probability $1 - \Psi_p \in [0, 1]$. When a firm does not change its price, it applies an indexation rule according to $P_t(h) = P_{t-1}(h)\Pi_{t-1}$. The optimal nominal price common to all firms is equal to

$$E_t\Sigma_{s=0}^{\infty}\Psi_p^j M_{t,t+s}Y_{t+s}\left[\left(\frac{\Pi_t}{\Pi_{t+s}}\right)^{(1-\theta_p)}\left(\frac{P_t*}{P_t}\right) - \frac{\theta_p}{\theta_p - 1}mc_{t+s}\left(\frac{\Pi_t}{\Pi_{t+s}}\right)^{-\theta_p}\right] = 0 \tag{33}$$

The law of evolution of price of final goods, $P_t$, and its dispersion $\Delta_{p,t}$ can be written as:

$$P_t = (1 - \Psi_p)(P_t*)^{(1-\theta_p)} + \Psi_p(\Pi_{t-1}P_{t-1})^{(1-\theta_p)}, \tag{34}$$

$$\Delta_{p,t} = (1 - \Psi_p)\left(\frac{P_t*}{P_t}\right)^{-\theta_p} + \Psi_p\left(\frac{\Pi_{t-1}}{\Pi_t}\right)^{-\theta_p}\Delta_{p,t-1}. \tag{35}$$

*3.3. Monetary Policy*

The central bank pursues monetary policy according to the Taylor rule:

$$\frac{I_t}{I} = \left(\frac{I_{t-1}}{I}\right)^{\Psi_I}\left(\frac{\Pi_t}{\Pi}\right)^{(1-\Psi_I)\Psi_\pi}\left(\frac{Y_t}{Y}\right)^{(1-\Psi_I)\Psi_Y}\omega_t^I, \tag{36}$$

where $I$, $\Pi$ and $Y$ are interest rates, inflation and output in steady states; $\Psi_I$, $\Psi_Y$ and $\Psi_\Pi$ are smoothing parameters for interest rate, output and inflation in the Taylor rule, and $\omega_t^I$ is monetary policy shock and evolves according to an AR(1) process:

$$log\omega_t^I = \rho_\omega log\omega_{t-1}^I + \sigma_t^I \eta_t^I, \tag{37}$$

where $\rho_\omega$ and $\sigma_t^I$ are the persistence and time-varying volatility of monetary policy shocks, respectively. Time-varying volatility of monetary policy ($\sigma_t^I$) is supposed to follow an AR(1) process:

$$\sigma_t^I = (1 - \rho_{\sigma^I})\sigma^I + \rho_{\sigma^I}\sigma_{t-1}^I + \sigma^{\sigma^I}\epsilon_t^{\sigma^I}, \tag{38}$$

where $\epsilon_t^{\sigma^I}$ is monetary policy uncertainty shocks.

*3.4. Market Clearing Conditions*

Market clearing conditions are given by the following equations:
- Goods markets:

$$Y_t = C_t + X_t \tag{39}$$

- Asset Markets:

$$\int_0^1 B_t(j)dj = 0 \tag{40}$$

- Stock Markets:

$$\int_0^1 V_t(j,h)dj = 1 \text{ for all } h \in [0,1] \tag{41}$$

*3.5. COVID-19 and Uncertainty Shocks*

To reflect the change in FOMC monetary policy strategy as reflected in the 2020 statement on Longer-Run Goals and Monetary Policy Strategy, following the Federal Reserve Bank of New York DSGE model documentation, we replace the monetary policy rule in Equation (36) with the following interest rate reaction:

$$I_t = \Psi_I I_{t-1} + (1 - \Psi_I)(1 - \rho_p)\Psi_\pi pgap_t + (1 - \Psi_I)(1 - \rho_y)\Psi_y ygap_t + \omega_t^I \tag{42}$$

In Equation (42), the inflation gap and the output gap are computed as follows:

$$pgap_t = (\Pi_t - 2) + \rho_p pgap_{t-1} \tag{43}$$

$$ygap_t = (\Delta y_t + Z_t) + \rho_y ygap_{t-1} \tag{44}$$

In Equation (42), $\Pi_t$ is core PCE inflation and is calculated as follows:

$$Core\ PCE_t = \Pi_t + 100 * (\pi^* - 1) \tag{45}$$

Equation (42) suggests that the interest rate responds to a moving average of deviations of inflation from the FOMC long-term goal of 2%. Similarly, the central bank responds to the moving average of deviations of output from its steady state. Lastly, we rewrite the volatility equation as follows:

$$\sigma_t^I = (1 - \rho_{\sigma^I})\sigma^I + \rho_{\sigma^I}\sigma_{t-1}^I + \sigma^{\sigma^I}\epsilon_{t-1}^{\sigma^I}, \tag{46}$$

which, we believe, represents the situation in 2020-Q2 when people had already started anticipating uncertainty in policy.

### 3.6. Parameterization of the Model

The parameterization of the model is based on Fernandez-Villaverde et al. (2011), Christiano et al. (2014); Challe and Giannitsarou (2014). The household discount factor $\beta$ is set to be 0.99, which represents the steady-state value of interest rates to be 4 percent per year. The steady-state inflation ($\pi$) is set to 1.0045 in accordance with Leduc and Liu (2016), which represents the Federal Reserve's inflation objective. Frisch elasticity of labor supply $\nu$ is fixed at 1 so that the steady state value of labor (N) is calculated to be 0.33. Similarly, we set the value of disutility of the labor supply $\chi$ to be 0.564 to match the steady state value of the labor supply. The habit parameter, b, is set to 0.8 following the existing literature such as Leduc and Liu (2016). The Taylor rule parameters are set to $\Psi_\pi = 1.01$, $\Psi_y = 0.6$ and $\Psi_R = 0.9$, which are in line with the estimates from Smets and Wouters (2007). Finally, the parameters of the policy uncertainty process have been set $\rho_\sigma^I = 0.9$ and $100\sigma^{\sigma^I} = 1$.

The elasticities of the demand for goods and labor products are set to $\theta_p = \theta_w = 4$. Finally, we set the degree of price rigidity $\psi_p = 0.6$ and the degree of wage rigidity to be $\psi_w = 0.9$.

For COVID-19-related parameters, following the Federal Reserve Bank of New York guidelines, we set $\rho_p$ and $\rho_y$ to be equal to 0.93. Similarly, we set the value of pgap and ygap equal to $-0.125$ and $-12\%$ so that the role of uncertainty impacts on economic contractions could be studied. A pgap of $-0.125$ represents the average gap of inflation from the FOMC target of 2% over the last 5 years, while the value of ygap indicates the output contraction due to COVID-19-induced recessions in 2020-Q2. We set the value of interest rate persistence, $\rho_I$, to be 0.75. Additionally, we set the value of output and inflation feedback parameters, $\Psi_y$ and $\Psi_\Pi$ to be equal to one and three, respectively. These values neither represent the view of FOMC nor that of the Federal Reserve Bank of NY. We chose those values so that we could demonstrate the uncertainty impacts in the presence of new policy regimes. Based on the empirical model, we set the value of $\rho_{\sigma^I}$ to 0.9 and that of $\epsilon_t^\sigma$ to 1.19. All the other variables in the new model are the same as in the baseline model.

### 3.6.1. DSGE Interpretation of Empirical Results

Empirical results suggest that uncertainty shock impacts on the US economy depends upon the state of the economy. Most importantly, the empirical findings suggest that uncertainty impacts were the largest on GDP during COVID-19. In this subsection, we study uncertainty impacts on the model economy during COVID-19. Furthermore, we also perform a number of simulations by changing the value of structural parameters so that we obtain a better understanding of reasons behind state-dependent uncertainty impacts in normal times. Following Mumtaz and Theoridoris (2018), we prefer simulation to estimation as the nonlinear nature of uncertainty shocks in DSGE makes estimation of impulse response a complex and time-consuming task. Furthermore, Canova and Sala (2009) argue that the simulated method of moments used for uncertainty estimation might give biased results due to weak or partial identification.

### 3.6.2. COVID-19 and Uncertainty Shocks

Figure 6 below compares the monetary policy uncertainty shocks on the model economy with and without accounting for policy changes to COVID-19. The model economy without COVID-19 contains the standard Taylor rule according to Equation (36) and the uncertainty shock equation given by Equation (38).

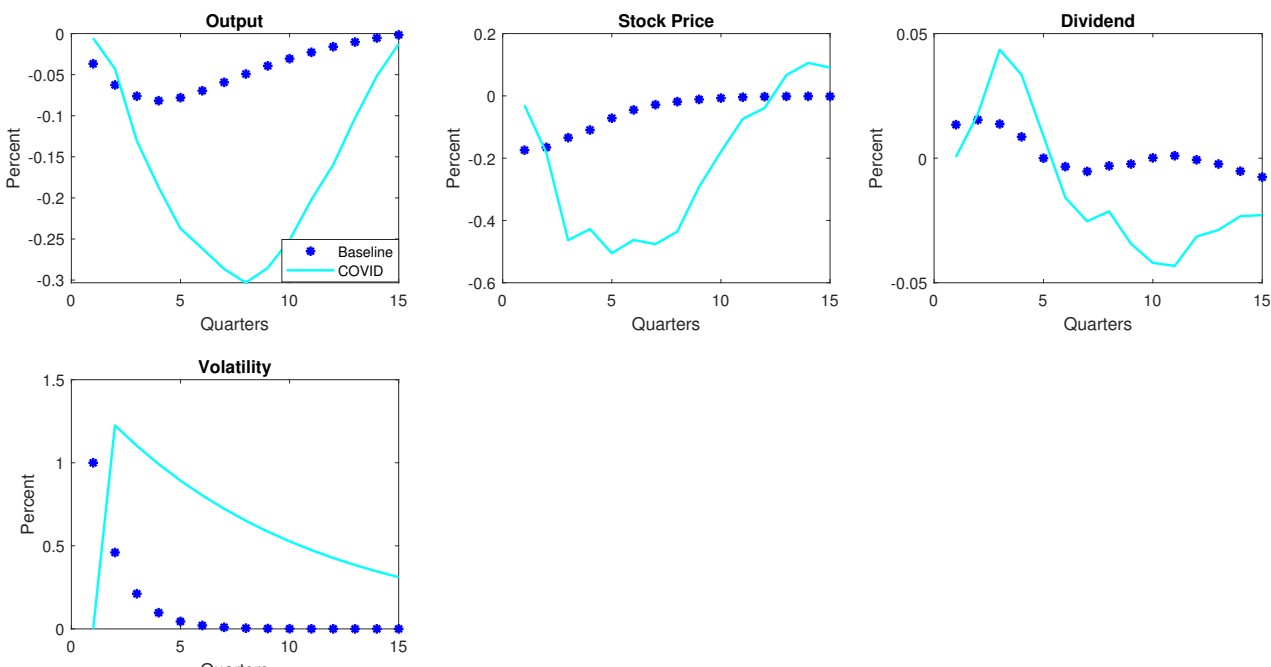

**Figure 6.** Policy uncertainty shocks. Notes: Figure 6 shows impacts of a one standard deviation increase in uncertainty in the baseline and in the presence of an alternate policy equation.

Figure 6 shows responses of macroeconomic and financial variables due to uncertainty shocks. The solid cyan line represents the response of variables when policy changes during COVID-19 are taken into account, while the cyan asterisks indicate uncertainty impacts in the baseline model. The figure shows that policy uncertainty caused persistent and larger negative impacts on output, stock prices and dividends in the presence of COVID-19 adjustments relative to the baseline model. This finding is in agreement with the empirical result that uncertainty impacts on output were largest during the COVID-19 recession compared to previous recessions during the sample.

The empirical findings suggest that apart from COVID-19, there are different phases in the US business cycle where uncertainty shocks impacts might have increased or decreased. There are many papers in the new Keynesian economics literature explaining the effects of changes in the policy reaction function (see, for example, Davig and Leeper 2007; Mumtaz and Theoridoris 2018). Similarly, a number of authors also studied changes in wage setting and price setting behavior (see, for instance, Hoffman et al. 2012). Therefore, it is imperative for us to find out whether the changes in those parameters can explain the dynamic nature of uncertainty shock impacts on the US economy. In the next subsection, we conduct a number of simulations to study the effects of changes in different parameters on uncertainty shocks.

### 3.6.3. Hawkish/Dovish Monetary Policy

We start by asking what happens if the central bank becomes hawkish and/or dovish in targeting inflation while formulating monetary policy. In Figure 7, the cyan line represents uncertainty impacts when the weight on inflation ($\Psi_\pi = 1.01$), and the blue dotted lines indicate the response when the central bank becomes hawkish on inflation targeting, i.e., $\Psi_\pi = 2.03$.

Figure 7 suggests that the magnitude of the effects of uncertainty shocks changes depending upon the hawkishness of the central bank on inflation. Specifically, increasing weight on inflation reduces uncertainty impacts on macroeconomic as well as on financial variables. When $\Psi_\pi$ increases, inflation will be right on its way to meeting the targeted value and reducing households' and firms' concerns regarding future price movements, which dents uncertainty impacts overall.

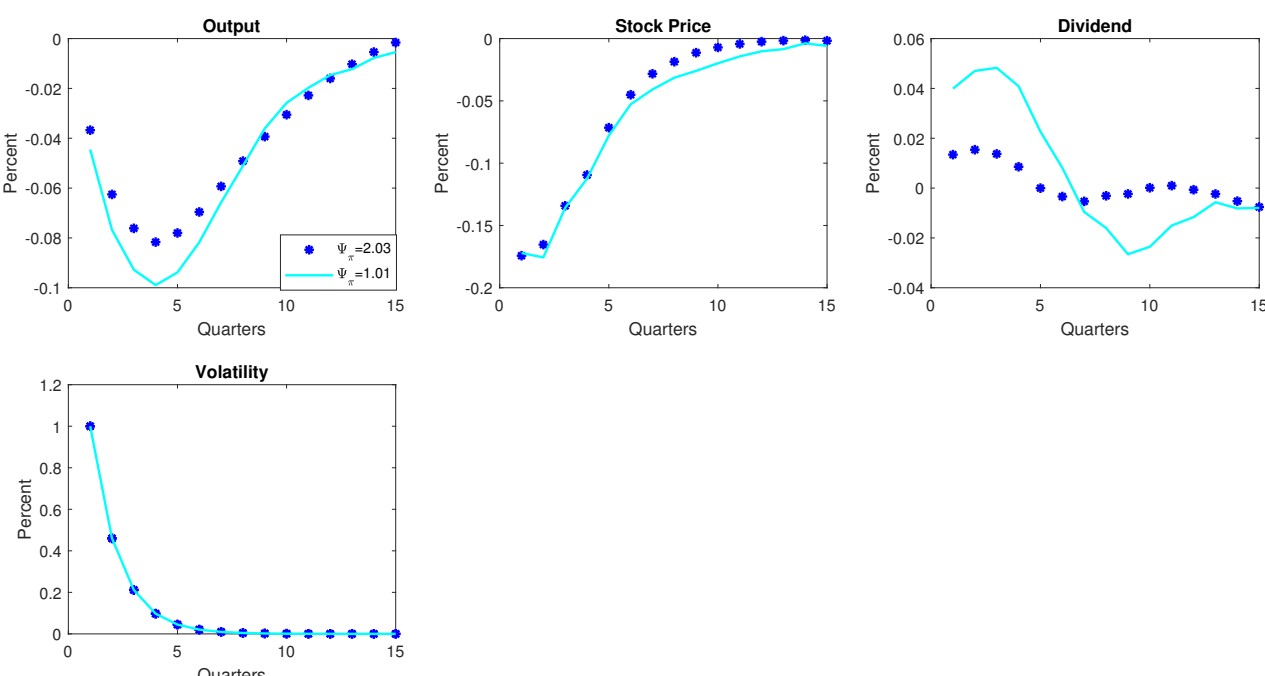

**Figure 7.** Hawkish/Dovish central bank. *Notes*: Figure 7 shows the impulse response of variables to uncertainty shocks. The solid line represents when the inflation weight on monetary policy is $\Psi_\pi = 1.01$ and the asterisks show the response when the central bank becomes hawkish, i.e., $\Psi_\pi = 2.03$.

### 3.6.4. Price and Wage Rigidity

Now, we study the impacts of uncertainty shocks when rigidities in price and wages are reconsidered. Figure 8 shows uncertainty impacts when parameters corresponding to price and wage stickiness are changed. For instance, PWR in the figure represents the response of variables when both price and wages are sticky according to the Calvo pricing mechanism. This simulation assumes the value of the Calvo probability of resetting wages, $\psi_w$, and the wage indexation parameter, $\theta_w$, to be equal to 0.9 and 1, respectively. Similarly, we assume the value of the Calvo probability of resetting price, $\psi_p$, and the price indexation parameter, $\theta_p$, to be equal to 0.6 and 1, respectively. PWF indicates the uncertainty shock response when both price and wages are flexible. This simulation assumes the value of $\psi_w$ and $\psi_p$ to be equal to 0.05, and $\theta_w$ and $\theta_p$ are considered to be equal to be zero. WR refers to the uncertainty shock response when only the wage is rigid, and the price is flexible. This simulation assumes the value of $\theta_w$ and $\psi_w$ to be equal to 0.9 and 1, respectively, while $\theta_p$ and $\psi_p$ take the values of 0 and 0.05. Lastly, PR refers to responses when the price is rigid, and the wage is flexible. This simulation assumes the value of $\theta_w$ and $\psi_w$ to be equal to 0 and 0.05, respectively, while the values of $\theta_p$ and $\psi_p$ are fixed to 1 and 0.6, respectively.

Figure 8 shows that wage and price rigidities amplify the negative impacts of uncertainty shocks on macroeconomic and financial variables. In fact, uncertainty impacts becomes more persistent and larger in the present of rigidities in price and wage compared to when both variables are allowed to change freely. The finding is in line with the literature in uncertainty shocks. The figure also provides an important insight into the relative importance of wage and price rigidity on the propagation of uncertainty shocks. It appears that price rigidity plays a dominant role in the negative impact of uncertainty shocks compared to wage rigidity. This is evinced by the behavior of impulse response functions when rigidities are considered separately. For instance, output decrease in the presence of price rigidity is almost similar to that due to rigidity in both price and wage. However, when only rigidity in wage is considered, the impact remains very small. Differing impacts of wage and price rigidities observed in Figure 8 suggest the relative importance of stickiness on uncertainty shock propagation.

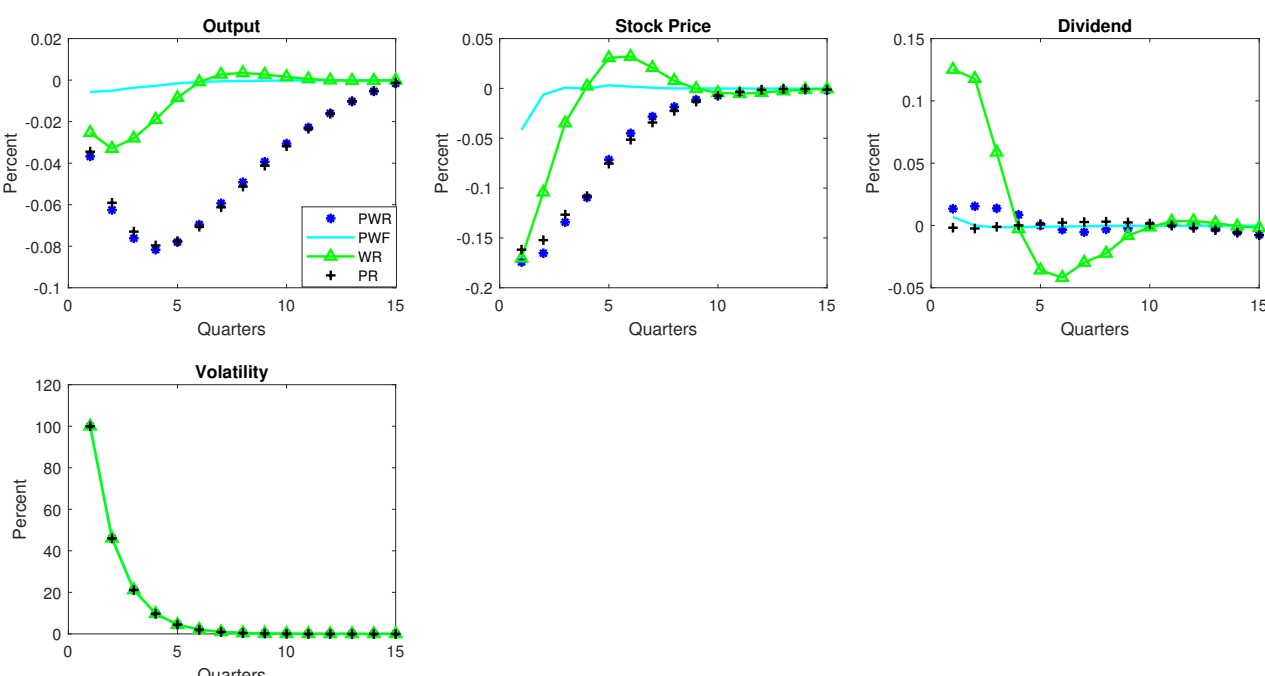

**Figure 8.** Price and wage rigidity and uncertainty impacts. *Notes*: Figure 8 shows the impulse response of variables to uncertainty shocks when parameters representing wage and price rigidities are changed. 'PWR' refers to responses when price and wage are both rigid. 'PWF' indicates responses in the presence of price and wage flexibility. 'WR' shows responses when only the wage is rigid, while 'PR' shows impacts when only the price is rigid.

## 4. Conclusions and Discussions

In this article, we assess whether the impacts of policy uncertainty shocks have changed over time. Using a Time-Varying Parameter Vector Autoregression (TVP-VAR), we find that uncertainty impacts are dynamic in nature. To be specific, the empirical analysis suggests that uncertainty impacts on output were largest during the COVID-19 recession compared to previous contractions. However, financial variables such as stock prices suffered the largest uncertainty impacts during the dot com bust of the late 1990s followed by that of the stock market rout in 2008/09. We use a nonlinear DSGE model framework to study possible factors behind the dynamic nature of uncertainty shocks. The model analysis suggests that rigidities in wage and price play a significant role in changing uncertainty propagation. Furthermore, we find that any change in the weight given to inflation by the Federal Reserve on monetary policy rules affects the extent of uncertainty propagation. Finally, we study the impacts of uncertainty during the COVID-19 recession by introducing the average inflation targeting rule in the DSGE model. Such an extension in the baseline model mimics the elevated uncertainty impacts on macroeconomic and financial variables observed in our empirical model. These findings suggest that uncertainty propagation depends upon monetary policy, inflation dynamics and the state of the economy.

**Author Contributions:** Conceptualization, A.K. and N.P. K.; methodology, A.K. and N.P.K.; software, A.K. and N.P.K.; validation, A.K. and N.P.K.; formal analysis, A.K. and N.P.K.; investigation, A.K. and N.P.K.; resources, A.K. and N.P.K.; data curation, A.K. and N.P.K.; writing—original draft preparation, A.K. and N.P.K.; writing—review and editing, A.K. and N.P.K.; visualization, A.K. and N.P.K.. All authors have read and agreed to the published version of the manuscript.

**Funding:** This research received no external funding.

**Informed Consent Statement:** Not applicable.

**Data Availability Statement:** Data is sourced from the websites of Robert Shiller, http://www.econ.yale.edu/shiller/data.htm (accessed on 5 September 2022); and Lucas Husted, https://sites.google.com/site/lucasfhusted/data (accessed on 7 September 2022).

**Conflicts of Interest:** The authors declare no conflict of interest.

## Appendix A

*Uncertainty Impacts Using the Michigan Uncertainty Proxy*

In this section, we use the fraction of respondents highlighting future uncertainty as a reason behind a household decision on an automobile purchase as the uncertainty proxy. The index has been increasingly used in the literature (see, for instance, Leduc and Liu 2016). Uncertainty impacts in this new framework are reported in Figures A1 and A2 below.

As in the baseline model, dynamic impacts of uncertainty shocks can be represented by three-dimensional impulse response functions as in Figure A1 below.

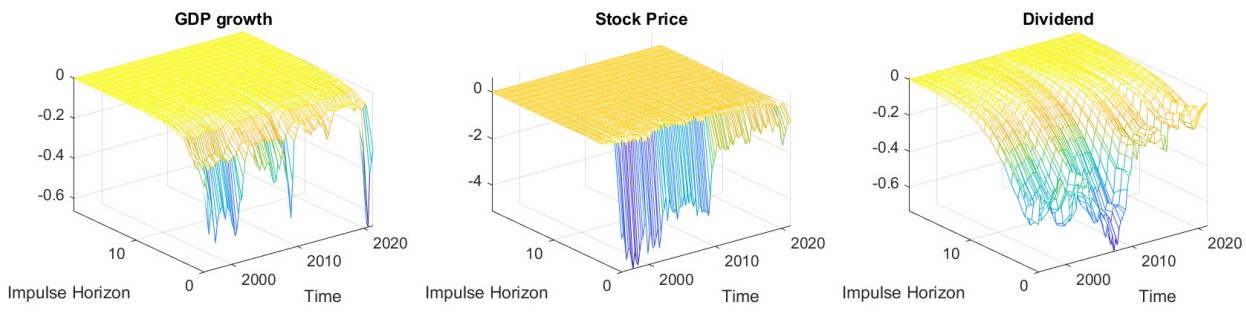

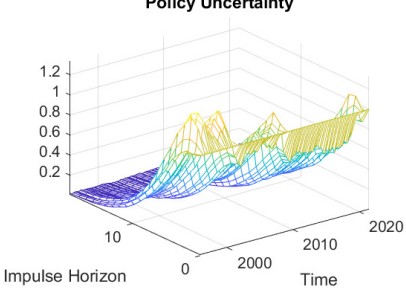

**Figure A1.** Impulse response functions to uncertainty shocks. *Notes*: Figure A1 shows one standard deviation of uncertainty.

Figure A1 shows the dynamic impacts of uncertainty shocks from 1995 to 2021. As in the benchmark model, uncertainty impacts on output growth are largest during the COVID-19-induced recession followed by the Great Recession of 2008/09. Similarly, stock prices appear to have been affected worst by uncertainty during the Asian crisis of the late 1990s followed by stock market routs during the Great Recession. Dividends also show a negative response following the rise in uncertainty and the largest negative impacts on dividends seem to have appeared during the Great Recession. Lastly, consumer uncertainty increases due to uncertainty rise, and the rise was the largest during the COVID-19-induced recession. One possible point of difference between the benchmark model and the new model is possibly the larger uncertainty impacts on output in the presence of Michigan Uncertainty as uncertainty proxy. The difference might indicate that policy uncertainty and consumer uncertainty have impacts of different magnitudes. Elevated impacts of demand side uncertainty observed in this section is in line with the literature (see, for instance, Basu and Bundick 2017).

As in the baseline model, we study uncertainty shocks impacts on the US economy at three dates: 1997, 2008 and 2015. Impacts are shown in Figure A2.

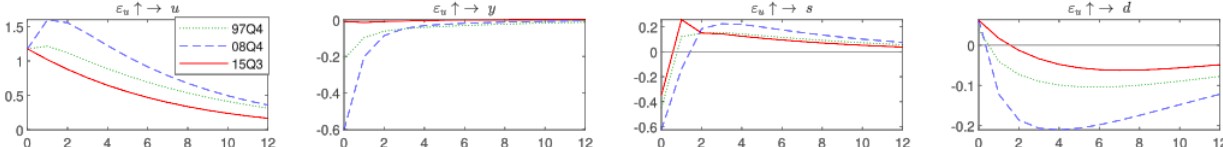

**Figure A2.** The Effects of Uncertainty Shocks in 1997, 2008, and 2015.　*Notes*: Figure A2 shows impulse response of variables to uncertainty shocks. - - indicates responses of variables to one standard deviation uncertainty shocks in 2015Q3. - - is response in 2008Q4 and the thin transparent line is response in 2015Q3. In the figure, 'u' means uncertainty, 'y' is gdp growth, 's' is stock price return, and 'd' means dividend.

Figure A2 shows the impacts of uncertainty shocks in 1997Q4, 2008Q4 and 2015Q3. As in the baseline model, we can see that uncertainty shocks had persistent and larger negative impacts on macroeconomic as well as financial variables during the Great Recession of 2008/09 followed by negative impacts during the Asian crisis of the late 1990s.

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
