# Peer review of "The Dynamic Impacts of Monetary Policy Uncertainty Shocks"

_economies, doi:10.3390/economies11010017_

Round 1

Reviewer 1 Report

The idea and its implementation is appropriate.  

Author Response

Thank you very much for reviewing and providing feedback on our work. 

Reviewer 2 Report

Major Comments

1) Uncertainty vs. Monetary Policy Uncertainty: The title of the paper and the abstract and introduction focus on broader notions of uncertainy, while the application and then the model focuses specifically on Monetary Policy Uncertainty (MPU). However, MPU has specifically been studied in Husted et. al. (2020) and is not the same as other forms of uncertainty. The paper needs to be much more clear as to why a specific form of uncertainty was chosen and why this matters. Why is MPU different the Economic Policy Uncertainty in this context, especially since the latter measure is available over a much longer sample? It is also not clear how the Michigan Consumer uncertainty relates to MPU and why this was chosen as an alternative proxy rather than other measures. The authors should clearly discuss why this would be related to be MPU.

2) VAR variables: Related to the above comment, the authors formulate a VAR of four variables: MPU, GDP, stock prices, and dividends. These variables differ from the VAR in Husted et. al. (2020) and the order differs in important ways as well. Why did the authors choose GDP and not industrial production since all of the other variables are financial variables which are observed at a high frequency? It would seem using a monthly VAR would make much more sense in this context. The authors need to do a much better job explaining their decisions and the implications that those have for the analysis.

3) Time-varying VAR: Looking at Figure 1 shows that there may be changes in variance going on in the MPU series. Why did the authors focus on a TVP-VAR and not a TVP-VAR with stochastic volatility? For example, the authors might want to consider the paper by Lenza and Primiceri (2022, JAE) on estimating a VAR after March 2020.

 4) Impulse Response Functions: The paper argues that uncertainty has a greater impact on the economy during COVID-19 than during other periods. However, when comparing different impulse responses across three different episodes in Figure 3, the paper completely ignores the 2020 time period. Given the focus on COVID-19 in the paper it is curious why this was excluded. The authors need to explain or discuss differences across the different time periods, particularly for dividends and also the differences for the alternative proxy of uncertainty.

Minor Comments

1) The figures are organized in such a way that makes it very difficult interpret them. All figures in panels of 4 should be portrayed in 2 rows and 2 columns and not randomly across 4 columns or 3 columns and 2 rows. This makes it very hard on the reader to compare.

Reviewer 3 Report

The paper tries to analyze the impact of uncertainty on the U.S. Economy estimating a time-varying VAR model. The authors give a high weight to the monetary policy uncertainty in their model. Other sources of uncertainty have a lower weight.

Page 1, line 32: It should be mentioned that during the COVID-19 crisis the central bank conducted a different monetary policy. Moreover, the fiscal policy was much more active.

Page 3, line 117: The four variable system should be better explained. It includes two financial variables and only macro variable, e.g. GDP. In the United States the dividends aren’t very important from a macroeconomic perspective. Especially if we note the huge amount of share buyback. I would suggest to include inflation rate or data describing the labor market.

Page 4, paragraph 1: Please give some results regarding the stationary properties of the series.

Page 4, Figure 1: I cannot see a good fit in the 2008/2009 recession. Moreover, uncertainty is very high in the beginning of 2003 and between 2016 and 2017.

Page 4, line 140: Please present the model specification of the VAR. How many lags are used? What is the specification of the deterministic part? Please present some diagnostic statistics.

Page 5: Please present only two graphs in one row. This is true for the other figures.

Page 5, second line: Please change Figure (1) into Figure (2).

Page 5, line 172: The dot com bubble burst in 2000. In 1997 Q4 we had the Asian crisis.

Page 10, equation (26): The equation is not readable. The end of the equation is missing.

Round 2

Reviewer 2 Report

The revisions to the paper raise more questions than they solve. The paper now focuses on the impact of monetary policy uncertainty without controlling for other aspects of uncertainty that could be correlated with the measure of interest and changing at the same time. The paper also shows that one is able to obtain the same results regardless of whether one uses monetary policy uncertainty, economic policy uncertainty, or consumers measures of uncertainty. This raises pretty fundamental questions about what the VAR is actually identifying as as uncertainty shock. Given the high correlation between alternative measures of uncertainty how can we say anything specific about the role of monetary policy uncertainty? 

Thus, the estimated VAR does not appear to be identified in any sense of the word since it is unclear what is actually driving the time-variation.  This then raises serious doubts about the comparison with the DSGE model itself which only allows for monetary policy uncertainty and thereby the results are completely dependent on this assumption since uncertainty only comes in to the model through the monetary policy function. 

The authors note in their response that they are focused not on forecasting but rather on the parameters and the dynamic effects of uncertainty on monetary policy. However, the authors should be aware that there is a very close link between forecasts and impulse responses see Plagborg-Moeller and Wolf (2021, ECTA) for example. Thus, to have any trust in the impulse responses that are being estimated one has to have a reasonable model and that is very much in doubt in this paper. 

Author Response

Dear reviewer,

Please find our response to your questions regarding our paper.

Best,

The Authors
